Sea-level rise; coastal risk; climate adaptation; collaboration; early warning systems

**Corresponding author:**
Tim Henri Josephus Hermans;
Email: t.h.j.hermans@uu.nl

# An integrated view on the uncertainties of sea-level rise, hazards and impacts, and adaptation

Tim Henri Josephus Hermans[1] , Renske de Winter[2], Joep Storms[3], Frances E. Dunn[4], Renske Gelderloos[5], Ferdinand Diermanse[2], Toon Haer[6], Dewi Le Bars[7], Marjolijn Haasnoot[2,4], Ymkje Huismans[2,5], Loes M. Kreemers[8], Eveline C. van der Linden[7], Stuart G. Pearson[5], Roelof Rietbroek[9], Aimee B.A. Slangen[4,10] , Kathelijne M. Wijnberg[11], Gundula Winter[2] and Roderik S.W. van de Wal[1,4,7]

[1]Institute for Marine and Atmospheric Research Utrecht, Utrecht University, Utrecht, The Netherlands; [2]Deltares, Delft, The Netherlands; [3]Department of Geosciences and Engineering, Delft University of Technology, Delft, The Netherlands; [4]Department of Physical Geography, Utrecht University, Utrecht, The Netherlands; [5]Department of Hydraulic Engineering, Delft University of Technology, Delft, The Netherlands; [6]Institute for Environmental Studies (IVM), Vrije Universiteit Amsterdam, Amsterdam, The Netherlands; [7]Royal Netherlands Meteorological Institute (KNMI), De Bilt, The Netherlands; [8]Psychology for Sustainable Cities, Knowledge Centre Society and Law, Amsterdam University of Applied Science, Amsterdam, The Netherlands; [9]ITC Faculty of Geo-Information Science and Earth Observation, University of Twente, Enschede, The Netherlands; [10]Department of Estuarine and Delta Systems, NIOZ Royal Netherlands Institute for Sea Research, Yerseke, The Netherlands and [11]Department of Civil Engineering and Management, University of Twente, Enschede, The Netherlands

## Abstract

While adapting to future sea-level rise (SLR) and its hazards and impacts is a multidisciplinary challenge, the interaction of scientists across different research fields, and with practitioners, is limited. To stimulate collaboration and develop a common research agenda, a workshop held in June 2024 gathered 22 scientists and policymakers working in the Netherlands. Participants discussed the interacting uncertainties across three different research fields: sea-level projections, hazards and impacts, and adaptation. Here, we present our view on the most important uncertainties within each field and the feasibility of managing and reducing those uncertainties. We find that enhanced collaboration is urgently needed to prioritize uncertainty reductions, manage expectations and increase the relevance of science to adaptation planning. Furthermore, we argue that in the coming decades, significant uncertainties will remain or newly arise in each research field and that rapidly accelerating SLR will remain a possibility. Therefore, we recommend investigating the extent to which early warning systems can help policymakers as a tool to make timely decisions under remaining uncertainties, in both the Netherlands and other coastal areas. Crucially, this will require viewing SLR, its hazards and impacts, and adaptation as a whole.

## Impact statement

Due to the existential threat that sea-level rise (SLR) poses to the Netherlands, scientists in the Netherlands study a wide range of topics related to adaptation to future SLR. However, we observe that collaborations between these scientists, and between scientists and policymakers, are limited. The novel contribution of this paper is therefore that it brought together a diverse group of scientists and policymakers in the Netherlands to develop a joint view on the most important uncertainties of SLR, hazards and impacts, and adaptation, set a common research agenda for their reduction where possible and discuss adaptation decision-making under remaining uncertainties. This is impactful because it allowed us to identify those uncertainties most relevant to adaptation decision-making and to align the expectations of scientists and policymakers. We find that collaboration across research fields is important to better communicate and reduce relevant uncertainties, and we discuss several opportunities for doing so. Another important conclusion of our paper is that some significant uncertainties, as well as the potential for large and rapidly accelerating SLR resulting from instabilities in the climate system, will remain in the coming decades. This message is particularly impactful for policymakers and raises the need for tools like early warning systems to plan adaptation under remaining uncertainties. We argue that to develop effective early warning systems, an integrated view on the uncertainties of SLR, hazards and impacts, and adaptation is crucial. Specifically, we recommend investigating whether meaningful early warning signals can be derived for major instabilities in the climate system and studying potential institutional and social responses to early warning signals. While our conclusions are based on the Dutch context, they also hold value for other coastal nations.

## Introduction

Sea-level rise (SLR) has major consequences for the Netherlands, such as increased flood risk, loss of intertidal areas, coastline retreat and saltwater intrusion (e.g., Oude-Essink et al., 2010; Wang et al., 2018; Haasnoot et al., 2020a; Van de Wal et al., 2024). Adaptation is therefore necessary, but planning adaptation is complicated by the large uncertainties in future SLR, the response of coastal systems to SLR and other changes, and the socioeconomic, institutional and political context in which adaptation decisions need to be made. Because of the potential for large and accelerating SLR (Fox-Kemper et al., 2021; Van de Wal et al., 2022), the Netherlands adopts a flexible adaptation plan that can be adjusted in response to scientific, physical or socioeconomic developments.

New scientific insights can form an important reason to adjust the adaptation strategy. Because the consequences of SLR are so important for the Netherlands, scientists in the Netherlands are studying a diverse range of topics related to SLR (see Supplementary Table S1 for a representative selection of recent studies). We divided these topics into three research fields (Figure 1a): (1) projections of changes in sea level, including their underlying processes; (2) the changes in coastal morphology, hydrology and salt intrusion that SLR contributes to and the resulting hazards and impacts; and (3) adaptation to SLR, including the conceptualization and evaluation of adaptation strategies, behavior, adaptation capacity and limits, and decision-making under uncertainty. Many of these topics are also relevant to other coastal regions, although their relative importance may vary depending on geographical, political, socioeconomic and other aspects. Each research field in Figure 1a involves different scientific disciplines, such as geoscience, engineering, ecology, economics and social and political science.

While the research fields in Figure 1a are interconnected, we observe that the interaction between scientists from different research communities is limited. For instance, in the Netherlands, research on SLR and hazards and impacts is typically presented at separate annual conferences (the Dutch Geoscience Conference and the Netherlands Centre for Coastal Research Days, respectively). A similar national conference on adaptation does not exist. Furthermore, these conferences are not regularly attended by policymakers and industry representatives, while policy-oriented events such as the 'National Day of the Sea-Level Rise Knowledge Programme' and the 'Dutch Delta Congress' are only sparsely attended by scientists. Although some scientists engage with non-academic organizations and institutions in projects such as the National Adaptation Strategy and the Dutch Climate Research Initiative, the connection between scientists and non-scientists needs to be strengthened.

Due to the limited interaction across research fields and between scientists and practitioners, different ideas exist about the importance and feasibility of uncertainty reductions. However, sharing information from one field could steer the research in other fields and make research more effective and beneficial for society. Without considering the information at the interface between different research fields and the information needs of practitioners, science cannot optimally inform adaptation decisions (Hewitt et al., 2017a; Hinkel et al., 2019; Kopp et al., 2019; Magnan et al., 2022; Van den Hurk et al., 2022; Hirschfeld et al., 2024, McInnes et al., 2024). For instance, the uncertainty tolerance of a specific user strongly influences which studies, uncertainties and processes should be considered to develop relevant sea-level projections (Hinkel et al., 2019).

To stimulate collaboration and define a common research agenda, a one-day workshop with 22 scientists and policymakers was held in June 2024 (see Supplementary Table S2 for a list of the participants and their expertise). The program of the workshop revolved around several break-out discussions, both between experts from the same research field and between experts from different research fields (Figure 1b). The break-out discussions allowed us to identify the most urgent needs for uncertainty reductions according to each of the three research fields in Figure 1a, and to contrast those needs with the feasibility of uncertainty reductions according to the other fields.

In this paper, we further develop and share the main ideas that emerged during the workshop, substantiated by a review of relevant literature. We draw from literature and professional experience to motivate our shared view on the main uncertainties identified in each research field, the scope for reducing those uncertainties and how each research field can benefit from enhanced collaboration (sections 'Uncertainties in sea-level projections', 'Main uncertainties' and 'Uncertainties in adaptation'). Additionally, because we find that each research field has important uncertainties that will likely not be resolved in the short term, we recommend several steps to investigate the extent to which early warning systems can support adaptation

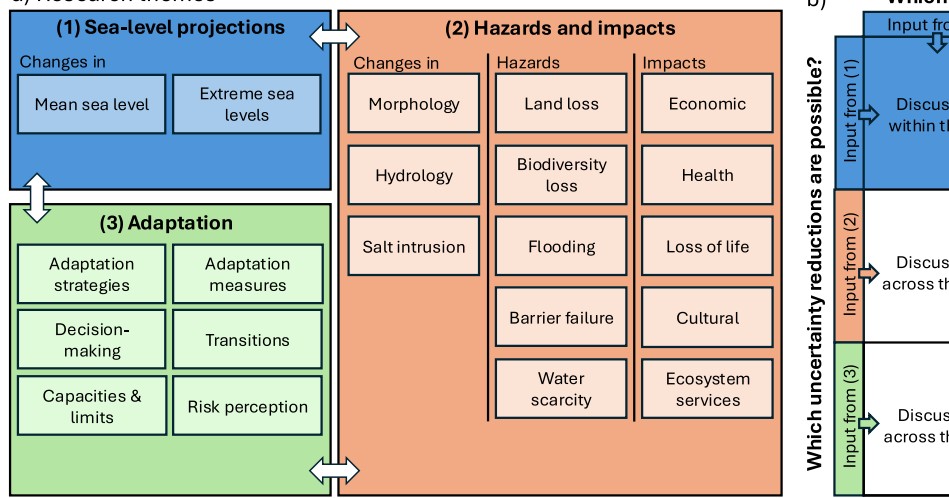

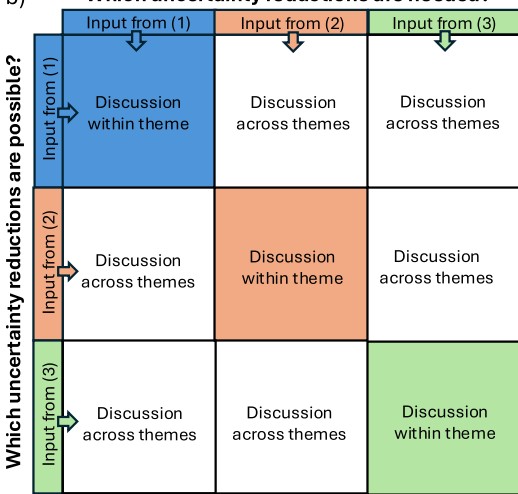

**Figure 1.** (a) Overview of research within the sea-level projections, hazards and impacts, and adaptation fields (adapted from Figure 4.1 of Oppenheimer et al., 2019). (b) Schematic illustration of the break-out discussions held during the workshop to identify desired and possible uncertainty reductions within and across different research fields.

decision-making under remaining uncertainties (section 'Toward effective early warning systems').

## Uncertainties in sea-level projections

Sea levels are projected to change due to a regionally varying combination of processes (Fox-Kemper et al., 2021). In the Netherlands, thermal expansion, ocean dynamic changes and the melt of the Antarctic Ice Sheet contribute the most to mean SLR. The additive effect of mean SLR has a large influence on the height of short-lived, extreme sea levels (e.g., Fox-Kemper et al., 2021; Hermans et al., 2023). In comparison, future changes in extreme sea levels in the Netherlands due to atmospheric changes are thought to be small (van Dorland et al., 2024). Therefore, we focus on projections of mean SLR in this section.

### *Main uncertainties*

We identify three categories of uncertainties in projections of mean SLR (Figure 2). Category 1 includes uncertainties in future greenhouse gas emissions, as reflected by curves S1 (low emissions) and S2 (high emissions) in Figure 2. Under each scenario, the projected SLR has inherent but quantifiable uncertainty (blue and red shading; Category 2) due to differences between the models and parameterizations used, and internal climate variability. The uncertainty in Category 2 is often quantified by means of a probability distribution between specific percentile bounds (e.g., Fox-Kemper et al., 2021). Finally, projected SLR has deep uncertainty (Category 3), which cannot be quantified unambiguously because experts do not agree on the characterization of specific processes contributing to SLR (Fox-Kemper et al., 2021; Abram et al., 2019; Kopp et al., 2023).

Deep uncertainty can be included in high-impact, low-likelihood scenarios (e.g., Van de Wal et al., 2022) in which SLR may depart from a more likely trajectory and rapidly accelerate (dashed lines in Figure 2). Such departures in SLR can be associated with tipping points in the climate system, which refer to critical thresholds beyond which physical systems strongly change, typically abruptly and irreversibly (Chen et al., 2021). Relevant examples are the potential

collapse of the Atlantic Meridional Overturning Circulation (AMOC) and the West Antarctic Ice Sheet. In the Netherlands, these processes could, respectively, lead to almost a meter of SLR (Levermann et al., 2005; van Westen et al., 2024) and multiple meters of SLR (Fox-Kemper et al., 2021), although their timescales may differ. Additionally, in case of the collapse of specific Antarctic glaciers, the accelerated trajectory of mean SLR could return to its original trajectory (see Figure 2) because only a finite amount of ice can be lost from a drainage basin.

Although these high-impact, low-likelihood scenarios are deeply uncertain, warnings of their materialization may be obtained by monitoring if departures of SLR from a more likely trajectory (Category 3) exceed the quantifiable uncertainty of that trajectory (Category 2) (e.g., Haasnoot et al., 2018; Stephens et al., 2018). This is indicated by the star in Figure 2. However, as indicated by the horizontal arrows in Figure 2, it may be possible to obtain earlier warning signals by monitoring potential precursors of crossing tipping points in addition to monitoring the accelerations in SLR it may cause. This will be discussed in more detail in the 'Toward effective early warning systems' section.

### *Scope for reducing uncertainties*

Uncertainty in future emissions (Category 1) will likely decrease over time because emissions scenarios will be more strongly constrained by longer records of historical greenhouse gas emissions, trends in the energy sector and pledges of and progress in mitigation (Hausfather and Peters, 2020). For instance, based on current policies and nationally determined contributions to emission reductions, it is unlikely that global warming will be limited to 1.5 degrees without a strong overshoot (United Nations Environment Programme, 2024). However, the dependence of climate tipping on warming is poorly constrained (McKay et al., 2022), and some tipping points relevant to SLR may have already been crossed. For example, recent studies suggest that the AMOC is already on a tipping course (van Westen et al., 2024) and that the collapse of the Thwaites and Pine Island Glaciers in West Antarctica will occur regardless of further increases in greenhouse gas concentrations (Van den Akker et al., 2025). Therefore, stronger constraints on future emissions do not necessarily rule out the potential for large and rapidly accelerating SLR.

With continued and new observations, we expect that quantifiable uncertainties of SLR (Category 2) can partially be reduced. For instance, emergent constraints may be used to reduce the spread between climate models (e.g., Lyu et al., 2021; Le Bars et al., 2024), ice discharge observations to improve basal melt parameterizations (Van der Linden et al., 2023) and observations of ice-shelf cavities to improve process understanding (Rignot, 2023; Vankova et al., 2023). More observations, increasing paleoclimatic evidence and continued model development may also help partially quantify and/or reduce deep uncertainty (e.g., Morlighem et al., 2024). However, this may also reveal new surprises and additional 'unknown unknowns' that will increase deep uncertainty instead of reducing it (Kopp et al., 2019).

Alongside the importance of observations, two key model developments are needed to better understand the deeply uncertain potential for large and rapidly accelerating SLR in The Netherlands (Category 3). First, global climate models with a higher spatial resolution are needed to better evaluate the potential slowdown and/or reversal of the AMOC (Hirschi et al., 2020) and its consequences for SLR in the Netherlands (Holt et al., 2017; Wise et al., 2024). This is important to pursue because by explicitly representing mesoscale eddies and simulating more realistic boundary

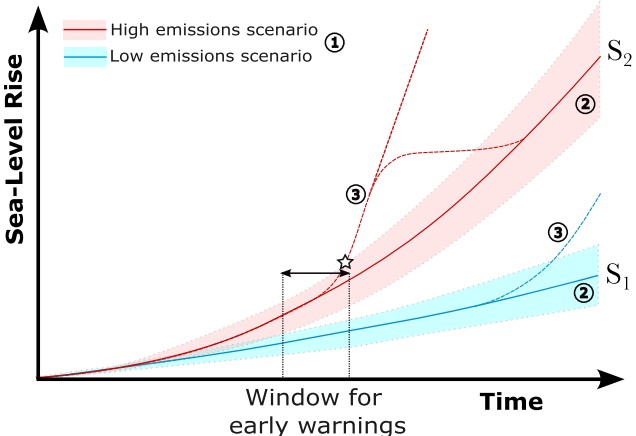

**Figure 2.** Schematic projections of mean SLR for a low (S1, blue) and high (S2, red) emissions scenario (Category 1). The shading around S1 and S2 depicts quantifiable uncertainty (Category 2). The dashed lines represent deep uncertainty (Category 3) related to tipping behavior that may lead to a (temporary) departure of mean SLR from S1 or S2. The star indicates when such a departure may emerge from the quantifiable uncertainty of the projections, and the black arrow represents the time window during which early warnings of this departure may be received.

currents, high-resolution models are providing new scientific insights into the AMOC and may shed more light on its potential bistability (Hewitt et al., 2017b; Hirschi et al., 2020). Increased spatial resolution is also important for simulating the influence of changes in the Southern Ocean circulation on the basal melt of the Antarctic Ice Sheet (van Westen and Dijkstra, 2021) and the effect of ocean and shelf sea dynamics on coastal sea-level change (Jevrejeva et al., 2024).

However, century-long simulations with global climate models at kilometer-scale resolution have high computational costs and storage demands (e.g., Van Westen et al., 2021). Therefore, routinely running them will likely remain uncommon in the coming decades (Holt et al., 2017; Jevrejeva et al., 2024). Augmenting coarse-resolution simulations with data-driven parameterizations learned from high-resolution simulations may help accelerate improving the representation of small-scale processes in climate models (Eyring et al., 2024).

Second, coupling between ocean- and ice-sheet models, in tandem with further separate model development, is urgently needed to represent the ice-ocean feedbacks that are typically absent in global climate models (Golledge et al., 2019), and to quantify their effects on SLR. Coupling is particularly important for the Southern Ocean-Antarctic Ice Sheet system, where ice loss is dominated by basal melting of ice shelves by a warm(ing) ocean. While the coupling of ocean and ice sheet models is gearing up (e.g., Smith et al., 2021; Lambert et al., 2023; Park et al., 2023), we do not expect this to be a common feature of global climate models in the Coupled Model Intercomparison Project 7.

### Importance of collaboration to produce actionable information

To communicate the complex uncertainties of sea-level projections described above and prioritize potential uncertainty reductions based on their relevance for impacts and adaptation planning, we find that collaboration with other research fields, and with practitioners, is crucial. Below, we provide two examples of how research on hazards and impacts (section 'Uncertainties in estimated hazards and impacts') and adaptation (section 'Uncertainties in adaptation'), and the user perspective of practitioners, can contribute to more actionable sea-level projections.

First, the practical relevance of sea-level projections can be increased by incorporating information on critical magnitudes, rates or timescales of SLR in their development. For instance, projecting increases in the exceedance frequency of water levels critical for the maintenance or closure of a storm surge barrier helps assess its remaining lifetime (Haasnoot et al., 2020a; Trace-Kleeberg et al., 2023). Similarly, the survival of intertidal flats and marshes can be assessed by projecting when critical rates of SLR may occur (Kirwan et al., 2016; Wang et al., 2018; Huismans et al., 2022). Furthermore, by incorporating existing levels of flood protection in projections of extreme sea levels (Hermans et al., 2023) and projecting when SLR thresholds corresponding to 'adaptation tipping points' may be exceeded (see Kwadijk et al., 2010), information can be obtained on the viability of existing water management strategies. Crucially, the relative importance of (reducing) the uncertainties identified in the 'Uncertainties in sea-level projections' section may vary in each of these cases because the uncertainties depend on the associated timescales (see Figure 2 and Slangen et al., 2022).

Second, the user perspective of practitioners is crucial to inform the development of sea-level projections, and of high-impact, low-

likelihood sea-level projections in particular (e.g., Katsman et al., 2011; Van de Wal et al., 2022). As discussed in the 'Uncertainties in sea-level projections' section, sea-level projections are typically conditioned on an emissions scenario and the tails of their probability distributions are difficult to quantify unambiguously. A key factor is therefore the risk tolerance of specific users, which determines the information that should be considered for the development of sea-level projections (Hinkel et al., 2019; Stammer et al., 2019). For a risk-tolerant user, for instance, considering SLR within uncertainty bounds in which experts have medium or higher confidence may suffice, while for more risk-averse users, information with a lower confidence level should be used (Hinkel et al., 2019).

Both of these examples underscore that the utility of sea-level projections depends on their users and the context of specific impacts and adaptation decisions (see also McInnes et al., 2024). This highlights the need for intensified inter- and transdisciplinary collaboration to produce more societally relevant sea-level science.

### Uncertainties in estimated hazards and impacts

SLR will impact the Netherlands substantially and in various ways. In this section, we discuss the uncertainty of changes in these hazards and impacts and highlight their interaction with the uncertainties of SLR (section 'Uncertainties in sea-level projections') and adaptation (section 'Uncertainties in adaptation') using three examples relevant to the Netherlands:

(1) *Increasing flood risk:* The flood-prone areas of the Netherlands (see Supplementary Figure S1) are protected by dunes, dikes and storm surge barriers. Without additional measures, SLR will increase flood risk (e.g., Aerts and Botzen, 2011; Paprotny et al., 2019; Tiggeloven et al., 2020; Haasnoot et al., 2020a), especially in low-lying areas along the coast and tidal rivers.

(2) *Retreat of the coastline and loss of intertidal areas:* Without increased nourishments, SLR will lead to more coastal erosion due to changes in sediment dynamics (e.g., Wang et al., 2018; Lodder et al., 2022). In estuaries and tidal basins, tidal flats and salt marshes may be lost in the long term if their vertical growth rate is outpaced by relative SLR and dikes prevent their inland migration (Pontee, 2013; Kirwan et al., 2016; Zhu et al., 2020; Huismans et al., 2022). This will degrade ecosystem health.

(3) *Salinization:* SLR increases salt intrusion in surface- and groundwaters in the Netherlands (Oude-Essink et al., 2010; Pauw et al., 2014; Van den Brink et al., 2019; Delsman et al., 2023; Van de Wal et al., 2024). This reduces the availability of freshwater for irrigation, drinking, sanitation, cooling and flushing polders and waterways (Mens et al., 2022), adversely affecting health, ecology, navigation and agriculture.

### Main uncertainties

The uncertainties in future SLR (see the 'Uncertainties in sea-level projections' section) translate into uncertainties in future hazards and impacts. However, we stress that moving down the impact chain, other sources of uncertainty also become important. For future flood risk, these concern, for instance, the influence of SLR and changes in coastal bathymetry on tides in shelf seas (Idier et al., 2017; Pickering et al., 2017) and river deltas (Leuven et al., 2023), and on storm surges and coastal waves (Arns et al., 2017). These hazards are, however, also influenced by human interventions, such as channel deepening and coastal management strategies (Idier

et al., 2017; Pickering et al., 2017; Leuven et al., 2023). Furthermore, future flood risk depends on the evolving standards, maintenance and (mal)functioning of flood defenses. For the Netherlands, the uncertainty of extreme sea levels is particularly relevant because the coastal flood protection standards in the Netherlands are associated with very low exceedance probabilities ($1/1,000$ yr$^{-1}$ to less than $1/10,000$ yr$^{-1}$), which are difficult to estimate (Van den Brink et al., 2005; Wahl et al., 2017).

Additionally, projected changes in flood risk are affected by uncertainties in (changes in) exposure and vulnerability (e.g., De Moel et al., 2011; Hinkel et al., 2021). Important factors are socio-economic developments, such as changes in land use and urban developments in flood-prone areas, societal dynamics (Aerts et al., 2018) and future adaptation measures (Tiggeloven et al., 2020). While additional flood protection measures may reduce flood risk, increasing the levels of flood protection may also promote investments in areas of residual risk, which then increases exposure and vulnerability (Di Baldassarre et al., 2018; Haer et al., 2020; Junger and Seher, 2024).

Enhanced retreat of the coastline and the potential loss of tidal flats, salt marshes and dunes critically depend on the rate of SLR versus the rates of (1) external sediment supply, from rivers, along-shore sources and the shoreface or shelf (de Winter and Ruessink, 2017; Bamunawala et al., 2021; Van der Spek et al., 2022; Lodder et al., 2023; Anthony et al., 2024; Aschenneller et al., 2024) and (2) internal sediment transport, to tidal flats, salt marshes, beaches and dunes (Van IJzendoorn et al., 2021; Huismans et al., 2022). Estimates of long-term sediment transport and morphological changes are subject to model uncertainties relating to unresolved or parameterized physical processes and assumptions of morpho-dynamic equilibrium (e.g., Becherer et al., 2018; Wang et al., 2018; Huismans et al., 2022; Lodder et al., 2022). Additionally, changes in morphology are heavily influenced by human interventions, such as nourishment, dam construction, dredging and mining (Elias et al., 2012; Siemes et al., 2024; Teixeira et al., 2024).

The largest uncertainty in future salt intrusion arises from climatic changes (e.g., Lee et al., 2025), namely changes in river discharge regimes following changes in precipitation and diminishing snowpacks (Rottler et al., 2021; Buitink et al., 2023), and SLR. Another source of uncertainty arises from modeling: 1D models, which are typically used to obtain long-term simulations (Mens et al., 2021), do not capture detailed salt dynamics well, while more accurate 3D models are too computationally expensive for long simulations. A third class of uncertainties arises from socioeconomic and climate-driven changes in water management and freshwater demand. For instance, groundwater recharge has a strong local dependency on precipitation and evaporation, land use and river and lake levels (Van Huijgevoort et al., 2020). While water infrastructure in the Netherlands is historically designed for optimal drainage of water to reduce the risk of flooding and facilitate farming, increasing salinization may necessitate a different approach (van der Brugge and de Winter, 2024; Vinke-de Kruijf et al., 2024a).

### Scope for reducing uncertainties

As discussed above, the uncertainties in future SLR (see the 'Uncertainties in sea-level projections' section) introduce uncertainty in future flood risk, coastal retreat and loss of intertidal areas, and salinization, but additional uncertainties arise from modeling these hazards and impacts, and their dependence on more direct human influences and other climatic changes. The scope for reducing the

uncertainty in SLR projections was discussed in the 'Scope for reducing uncertainties' subsection in the 'Uncertainties in sea-level projections' section. Regarding the uncertainty in projections of other relevant climate variables in the Netherlands, such as precipitation and temperature, we refer to van Dorland et al. (2024). As will be discussed in the next subsection, the incorporation of potential human interventions in projections of hazards and impacts affected by SLR requires considering future adaptation actions and other socioeconomic developments.

This leaves a discussion of potential reductions in model uncertainty. Like the uncertainty in SLR projections, the uncertainties in modeled future hazards and impacts may partially reduce over time with more observations, higher-resolution data and improved methods. For instance, the uncertainty in parameter estimates of extreme sea-level distributions can be reduced by exploiting spatial dependencies (e.g., Calafat and Marcos, 2020; Rashid et al., 2024). Additionally, more accurate digital terrain models are becoming available to model flooding (Pronk et al., 2024), although this is mainly relevant for regions less densely measured with Lidar than the Netherlands.

Similarly, uncertainties in modeling sediment transport and the associated coastal changes at decadal or longer timescales may be reduced by better resolving physical processes, such as waves and sand-mud dynamics (Huismans et al., 2022; Lodder et al., 2022; Colina Alonso et al., 2023), or by adopting reduced complexity (e.g., French et al., 2016; Reef et al., 2020; Portos-Amill et al., 2023), probabilistic (e.g., Keijsers et al., 2016; Toimil et al., 2017) and data assimilation approaches (e.g., Vitousek et al., 2017). To reduce uncertainty in modeling salt intrusion, a new method to combine a limited number of 3D simulations with long-term discharge statistics is being developed (Huismans et al., 2023). Other efforts to increase computational efficiencies, such as using width-averaged models with intermediate complexity, adaptive-sampling techniques and data-driven modeling, also provide new opportunities to reduce uncertainties in salt intrusion projections (Hendrickx et al., 2023; Wullems et al., 2023; Biemond et al., 2025).

### Importance of collaboration to produce actionable information

For flooding, coastal erosion and the fate of tidal flats and salt marshes, uncertainties in, specifically, the rate of SLR are most important to characterize and reduce where possible (see the 'Uncertainties in estimated hazards and impacts' section). To support impact assessments, sea-level projections could therefore more directly communicate future rates of SLR to users by explicitly presenting them in figures, instead of only including figures of SLR magnitudes from which rates need to be inferred (see Kopp et al., 2023, for a discussion on the role of figures as 'boundary objects' in climate-change communication). Additionally, (temporary) modulations of SLR rates by seasonal to multi-decadal variability and future changes therein (Widlansky et al., 2020; Hermans et al., 2022; Nandini-Weiss et al., 2024) may have relevant impacts but are typically not included in sea-level projections. We therefore argue that collaboration across research fields is needed to determine whether such underexposed changes are relevant and deserve more attention.

As exemplified in the 'Uncertainties in estimated hazards and impacts' section, future changes in hazards and impacts strongly depend on direct human influences that alter coastal and water systems and their physical responses, as well as exposure and vulnerability. However, these interactions are not always

considered. For instance, many flood risk assessments assume no or normative adaptation (e.g., Tiggeloven et al., 2020), which may lead to erroneous estimates of future flood risk. While future adaptation is difficult to project and the realization of adaptation strategies is uncertain itself (see section 'Uncertainties in adaptation'), adaptation scenarios that explore different adaptation options could be used to account for this uncertainty and illustrate the sensitivity of future flood risk to specific adaptation actions (Hinkel et al., 2021). For comprehensive flood-risk projections that integrate assessments of vulnerability and behavioral dynamics, multidisciplinary research is needed (Aerts et al., 2018; Haer et al., 2020).

Considering future interventions, such as changes in nourishment strategy, raising coastal defenses, freshwater use and relocation, is therefore crucial for projecting hazards and impacts. In other words, the hazards and impacts of future SLR need to be evaluated *in conjunction with* adaptation planning (section 'Uncertainties in adaptation'), rather than solely planning adaptation *in response to* hazard and impact assessments. Jointly determining the criteria for adaptation decisions will help identify which uncertainties in projected hazards and impacts are most critical and should be prioritized for reduction. This is supported by some of our practical experiences. For instance, complex and detailed salinization models may not be needed to evaluate potential freshwater intake locations for water boards if such locations can be rejected a priori based on local salinization risk tolerance and existing system knowledge. Similarly, a precise replication of salt concentration at the freshwater intake limit in models may not be worth pursuing, given the more significant uncertainties of industrial salt release upstream. As a final example, we observe that interdisciplinary discussions between modelers and ecologists in the Netherlands have been steering recent developments in sediment models.

## Uncertainties in adaptation

Under increasing hazards and impacts due to SLR (section 'Uncertainties in estimated hazards and impacts'), adaptation measures will be needed to keep the Dutch delta livable. This could entail increasing flood defenses and sand nourishment and preventing salt intrusions, through technological innovations and nature-based solutions, or more transformative, large-scale changes to land use and water infrastructure (e.g., Cooley et al., 2022; Haasnoot and Diermanse, 2022). As planning and implementation of coastal adaptation take time, decisions may need to be taken, while there is still large uncertainty in the projections of SLR (section 'Uncertainties in sea-level projections') and hazards and impacts (section 'Uncertainties in estimated hazards and impacts') (Haasnoot et al., 2020b; Glavovic et al., 2022). Moreover, adaptation decisions need to be made within a complex physical, cultural, socioeconomic, political-institutional and legal-governance decision space (Nicholls, 2018; Haasnoot et al., 2020b; Bongarts-Lebbe et al., 2021; Du et al., 2022; Vinke-de Kruijf et al., 2024a,b) and in the presence of other, increasing socioeconomic challenges and complexities, such as biodiversity loss, nitrogen emissions and housing shortage (Stokstad, 2019; Hochstenbach, 2024; Van der Brugge and De Winter, 2024).

### Main uncertainties

The Netherlands approaches adaptation flexibly using dynamic adaptive pathways (Haasnoot et al., 2013), involving cyclic assessments and monitoring (Haasnoot et al., 2018; Haasnoot et al., 2019; Van Alphen et al., 2022; Van der Brugge and De Winter, 2024). This results in a proactive and adaptive delta management strategy, which can provide effective adaptation strategies in the short and medium term. However, due to the large and deep uncertainties of SLR in the long term (see the 'Uncertainties in sea-level projections' section), no-regret measures based on likely scenarios of SLR are currently preferred. Large-scale transformations of the delta that are needed under multiple meters of SLR are currently being assessed as a far-future state (van Alphen et al., 2022) and are, to date, disconnected from present-day decisions and investments (ten Harmsen van der Beek et al., 2025). Without acknowledging long-term adaptation needs to SLR of potentially multiple meters (see the 'Uncertainties in sea-level projections' section), adaptation investments may result in maladaptation with lock-ins that are difficult or costly to further adapt from (Pörtner et al., 2022). For instance, investing in infrastructure in spaces that may later be needed for flood defenses or other measures reduces the solution space for adaptation.

Adaptation planning would benefit from reducing the uncertainties in the projections of SLR and the associated hazards and impacts (sections 'Uncertainties in sea-level projections' and 'Uncertainties in estimated hazards and impacts'). However, reductions in these uncertainties do not imply that (effective) adaptation will automatically follow, as challenging uncertainties of adaptation also sit within the social domain (O'Neill et al., 2022). For instance, limits to adaptation may arise from the path dependency of institutions and resistance to change (e.g., Barnett et al., 2015; Gupta et al., 2016), behavioral aspects such as risk perception, norms and efficacy beliefs (e.g., Van Valkengoed et al., 2022a; Van Valkengoed et al., 2024), poverty and social inequality and vulnerability (e.g., Tesselaar et al., 2020; Haer and de Ruiter, 2024; Vinke-de Kruijf et al., 2024b), political willingness and several other barriers and constraints (see Biesbroek et al., 2011; van der Brugge and Roosjen, 2015; Hinkel et al., 2018; O'Neill et al., 2022; Aerts et al., 2024).

A complicating factor is that the implementation of large-scale investments may need to start well before severe impacts on critical functions (e.g., agriculture, nature and housing) are experienced (ten Harmsen van der Beek et al., 2025). Additionally, preparing and implementing transformative decisions that fundamentally change the system (e.g., where and how people live) is societally challenging due to the difficulty of connecting short-term actions with long-term benefits, other political priorities and competing short-term economic decisions (e.g., Kates et al., 2012; van der Brugge and Roosjen, 2015; Coloff et al., 2021).

### Scope for reducing uncertainties

We highlight several needs and opportunities for reducing the uncertainties of adaptation discussed above. First, more research is needed to reduce the uncertainty of barriers and limits that may hinder effective adaptation, which currently have a sparse evidence base (Berrang-Ford et al., 2021; Berkhout and Dow, 2022; Juhola et al., 2024). Recent perspectives recommend studying, for instance, the empirical relationships between adaptation constraints and decisions and their integration in models, the temporal evolution of adaptation limits and the connection between adaptation limits and transformational adaptation (Berkhout and Dow, 2022; Lee et al., 2022; Aerts et al., 2024; Juhola et al., 2024). This also requires an improved understanding of the adaptive capacity of institutions (e.g., Gupta et al., 2016) and the motivating factors for and barriers to adaptation behavior by individuals and households (e.g., van Valkengoed and Steg, 2019; Van Valkengoed et al., 2022a; Sharpe and Steg, 2025), which are critical for designing effective adaptation interventions (Van Valkengoed et al., 2022b).

Second, further clarity is needed on the positive or negative interaction effects of (adaptation) decisions across time, space, sectors and actors (Dewulf et al., 2015; Challinor et al., 2018; Haer et al., 2020). For example, large-scale adaptation to SLR by governments can reduce the willingness of households to protect or insure. Furthermore, decisions to address other societal challenges, such as housing availability and other environmental pressures, can interfere or synergize with adaptation decisions to SLR (ten Harmsen van der Beek et al., 2025), and the changing political landscape and willingness of the population to adapt can significantly change the timeline of adaptation.

Finally, to address long-term impacts, the Netherlands might need to move from incremental adaptation, with a focus on preserving the present-day land use through measures like dike reinforcements and increasing pump capacity, to transformational adaptation, which fundamentally changes land use and spatial planning. Examples of the latter are large-scale land reclamation, closing off estuaries and river re-routing, and allowing low-lying areas to (occasionally) flood (Haasnoot et al., 2019; Kuhl et al., 2021). However, transformational adaptation involves increased costs, complexity and uncertainty. Research should therefore provide more clarity regarding the benefits of transformational adaptation, transition costs, timing and institutional and behavioral actions (Kates et al., 2012) and offer guidance on shaping adaptation pathways with positive future outlooks (Colloff et al., 2021; Haasnoot et al., 2024).

### Importance of collaboration to produce actionable information

During the workshop, policymakers expressed the need for clear communication of uncertainties and guidance on which scenarios and numbers to use. In this sense, adaptation planning can be supported by establishing pragmatic lower and upper bounds and the best projection of SLR and its hazards and impacts, based on transparent assumptions (Van der Brugge and De Winter, 2024; van Dorland et al., 2024). Doing so effectively sets a minimum, maximum and potentially most suitable adaptation path (Nicholls et al., 2021). Scientists can also support adaptation planners by expressing sea-level projections as the time when critical magnitudes or rates may be exceeded (Cooley et al., 2022; Slangen et al., 2022; Hermans et al., 2023), which helps to constrain the lead- and lifetimes of adaptation measures. Hazard and impact modeling is required to assess the consequences of implementing potential adaptation measures on coastal and ecological systems (see also subsection 'Importance of collaboration to produce actionable information' in the 'Uncertainties in estimated hazards and impacts' section).

Long-term and potentially transformational decisions are currently often stalled in anticipation of reductions in (deep) uncertainty in future SLR and its consequences (subsection 'Main uncertainties' in the 'Uncertainties in adaptation' section). However, predicting when and to what extent these uncertainties will be reduced is uncertain too, and not always possible (see sections 'Uncertainties in sea-level projections' and 'Uncertainties in estimated hazards and impacts'). Discussions during the workshop indicated that expectations of future uncertainty reductions were not well aligned between scientists from different research fields and decision-makers. This may lead to delayed adaptation in a time where swifter action is required and highlights the importance of continued conversations between these different groups.

In summary, we conclude from the previous sections that intensified collaboration across research fields and between scientists and practitioners is urgently needed to reduce decision-relevant uncertainties. Furthermore, significant uncertainties in each research field, and the potential to cross tipping points, will remain in a rapidly changing climate for decades or longer. To support robust decision-making under these uncertainties and minimize potential maladaptation, regret and lock-ins, decision-makers are advised to develop adaptive (pathway) plans (Haasnoot et al., 2013; Lempert, 2019). Monitoring the need for new decisions based on changing conditions is an integral part of such plans (e.g., Haasnoot et al., 2018), but the potential for obtaining early warnings of crossing the tipping points discussed in the 'Uncertainties in sea-level projections' section is not yet clear. In the 'Toward effective early warning systems' section, we therefore discuss the inter- and transdisciplinary research needed to investigate the potential for early warning systems as a novel component of adaptive plans.

### Toward effective early warning systems

An adaptive plan contains a monitoring component that defines which indicators to monitor and how and when signals triggering corrective policy or research actions could be derived (Haasnoot et al., 2013; Haasnoot et al., 2018). Such a monitoring system allows decision-makers to take near-term actions while keeping long-term options open and is therefore crucial for both designing and executing adaptive pathways plans (Haasnoot et al., 2013). Importantly, within a monitoring system, warning signals may arise from indications of changes in the uncertainties in each of the sea-level projections, hazards and impacts, and adaptation fields. For instance, indicators selected for the signal monitoring system of the Dutch Delta Programme include projected SLR (section 'Uncertainties in sea-level projections'), but also required volumes of sand nourishment, the frequency of storm surge barrier closures and impeded drainage (section 'Uncertainties in estimated hazards and impacts'), and changes in land use and population (section 'Uncertainties in adaptation') (Haasnoot et al., 2018). Furthermore, a combination of signals from different indicators may increase the value of those signals for decision-making.

Through monitoring the indicators selected by Haasnoot et al. (2018), indications that a climate tipping point has been crossed may be obtained when a rapid acceleration of SLR and/or its impacts emerges from quantifiable uncertainty (as marked by the star in Figure 2). However, by monitoring potential precursors of climate tipping points, earlier warnings may be obtained. We therefore propose investigating the potential for early warning systems to support adaptative plans.

For early warning systems to be effective, convincing signals need to be identified that (1) would lead to a substantial reduction in the uncertainty of relevant future changes in impacts and leave sufficient lead time for appropriate adaptation and (2) can be adequately acted upon by institutions and society. The multidisciplinary nature of these requirements strongly calls for an integrated view on sea-level projections, hazards and impacts, and adaptation. Therefore, we argue that collaboration across research fields is crucial to determine which (combinations of) early warning signals are actionable (Figure 3) and would lead to effective early warning systems. We recommend two specific directions of research in this regard, targeted at potential precursors of the climate tipping points that were discussed in the 'Uncertainties in sea-level projections' section (see section 'Investigate the use of precursors of instabilities

and tipping') and at the institutional and societal connectivity of early warning systems (see section 'Consider how early warning signals are used').

### *Investigate the use of precursors of instabilities and tipping*

Potential precursors of climate tipping points may serve as early warning signals. For instance, climate model simulations indicate that freshwater transport at 34°S is a precursor of AMOC collapse (Van Westen et al., 2024), and recent ice-sheet model simulations suggest that present-day mass loss rates are a precursor for the collapse of specific glaciers in West Antarctica (Van den Akker et al., 2025). More targeted simulations with climate- and ice-sheet models are needed to investigate these and other potential precursors of tipping points and instabilities that could be monitored, such as hydrofracturing on ice shelves and the temperature profiles in ocean cavities beneath ice shelves (Holland et al., 2020). Knowing the lead time between potential precursors and the crossing of tipping points is important to determine the value of precursors as a warning for necessary adaptation. However, the extent to which this lead time can be constrained, given current process understanding and model limitations, needs further investigation.

Additionally, we recommend exploring SLR and its hazards and impacts in what-if scenarios in which a collapse of the AMOC or (parts of) the West Antarctic Ice Sheet is imposed, using dedicated model experiments. By investigating whether the uncertainty in the consequences of instabilities can be sufficiently constrained, the value of potential early warning signals of those instabilities can be better assessed. For instance, while the timing of a collapse of the Thwaites and Pine Island glaciers in West Antarctica is uncertain, the rate of the resulting SLR appears to be relatively insensitive to parameter uncertainty (Van den Akker et al., 2025). If confirmed by other ice flow models, this could be used to constrain rate-dependent hazards and impacts (see the 'Uncertainties in estimated hazards and impacts' section) in such a scenario.

### *Consider how early warning signals are used*

Previous work has identified several criteria for an effective signal monitoring system for adaptation planning by governments (Haasnoot et al., 2018). However, the notion that the societal response to signals may also influence the effectiveness of governmental adaptation was not extensively considered. For example, businesses may interpret signals as a reason not to invest in low-lying areas and the flood-risk perception of citizens may influence the housing market (van Ginkel et al., 2022). Conversely, governmental flood protection can affect the incentive for adaptation by households (Haer et al., 2020). Therefore, the societal response to signals should also be considered in adaptive plans.

To ensure that early warning systems are valuable and actionable for policymakers, businesses and the public, and are integrated in decision procedures, co-designing them with their intended users is imperative (e.g., Hermans et al., 2017). If early warning signals do not reach the respective key decision-makers (in time) or will not be included in their decision-making, they will not yield their intended result. Psychological research on decision-making and adaptation responses shows how important response efficacy and decision context are in addition to plain knowledge or risk assessment (Van Valkengoed et al., 2024). Future research should therefore focus not only on obtaining the best signals, given the best available knowledge on SLR and hazards and impacts, but also on the best possible means to make the information conveyed by early warning signals available and relevant to key decision-makers, considering broader political contexts and connectivity to organizational decision-making (van der Steen and van Twist, 2012; Bossomworth et al., 2017). This requires a better understanding of where early warning signals may and should land, how and by whom they can be used and how they can be embedded in decision-making policy.

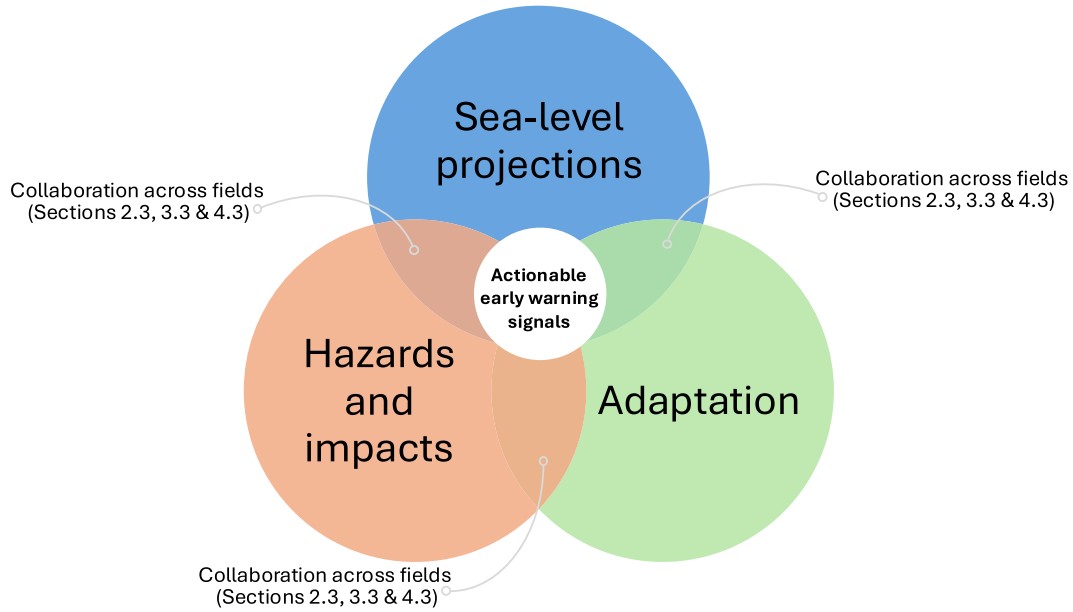

**Figure 3.** Schematic illustration of collaboration among the sea-level projections, hazards and impacts and adaptation fields, and the necessity of collaboration across fields to determine actionable early warning signals. Opportunities for collaboration are discussed in the 'Importance of collaboration to produce actionable information' subsection in the 'Uncertainties in sea-level projections', 'Uncertainties in estimated hazards and impacts' and 'Uncertainties in adaptation' sections.

## Conclusions

To conclude, we reiterate two main messages:

1. Intensified collaboration on SLR, its hazards and impacts, and adaptation is needed to set common research priorities, better align expectations of possible uncertainty reductions and increase the relevance of science to adaptation planning, as motivated in the 'Importance of collaboration to produce actionable information' subsection in the 'Uncertainties in sea-level projections', 'Uncertainties in estimated hazards and impacts' and 'Uncertainties in adaptation' sections. Addressing practical adaptation problems requires a holistic view on the chain of uncertainties across these research fields. Therefore, we recommend organizing conferences, events and/or platforms on SLR for broader audiences, enabling scientists from different fields, policymakers and industry to connect and discuss information needs in depth. We also stress the importance of both forming and funding multidisciplinary consortia to connect the ongoing work in the sea-level projections, hazards and impacts, and adaptation fields.

2. We anticipate that in the coming decades, significant uncertainties will continue to exist or arise in each of the SLR, hazards and impacts, and adaptation research fields, and that the potential for rapid changes in the climate system following instabilities and tipping points will remain. Therefore, we advise not to delay decision-making under the assumption that key uncertainties will be reduced in time, but to investigate the extent to which early warning systems can support timely decision-making in the presence of deep uncertainties that will remain. Crucially, identifying actionable early warning signals will require an integrated view on future SLR, its hazards and impacts, and adaptation.

Our view is based on the Dutch context, and we acknowledge that climate risks and the solution space for adaptation are region-specific. Nevertheless, different countries are facing similar adaptation challenges (e.g., Van den Hurk et al., 2022) and many of the uncertainties that we discussed are also relevant elsewhere. Therefore, we believe that our recommendations to intensify collaboration across research fields and between scientists and practitioners, and to further investigate the use of early warning systems, are also applicable to other coastal nations.

**Open peer review.** To view the open peer review materials for this article, please visit http://doi.org/10.1017/cft.2025.10003.

**Supplementary material.** The supplementary material for this article can be found at http://doi.org/10.1017/cft.2025.10003.

**Acknowledgements.** We would like to thank Hermine Erenstein, Annemiek Roeling (Ministry of Infrastructure and Water Management), Luc de Vries (Staff of the Delta Programme Commissioner) and Arjan Budding (Wageningen University and Research) for participating in the workshop and playing an advisory role in writing this paper. We would also like to thank Reint Jan Renes for his input on the manuscript revisions and Mariken van der Mark for supporting the organization of the workshop.

**Author contribution.** T.H.J.H. initiated and led the organization of the workshop and the writing of the manuscript. T.H.J.H., R.d.W., J.S. and F.E.D. conceptualized and organized the workshop and edited the manuscript (supported by R.S.W.v.d.W. and M.H.). R.G., F.D., T.H. and T.H.J.H. led the writing of the 'Uncertainties in sea-level projections', 'Uncertainties in estimated hazards and impacts', 'Uncertainties in adaptation' and 'Toward effective early warning systems' sections. All authors participated in the workshop and contributed to the manuscript.

**Financial support.** The workshop was funded by the Dutch Polar Program DP4C consortium (NWO).

**Competing interests.** The authors declare no competing interests.

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
