## [Reviewer Report]

This paper argues that significant uncertainties regarding sea-level rise, its impacts, and adaptation strategies will persist in the coming decades. It highlights the utility of early warning systems in supporting adaptive adaptation, which is a valuable and timely message. However, if the paper is intended to be published as a review, I must point out several issues. The methodology underpinning the review is not clearly presented, and the discussions in Sections 2, 3, and 4 lack depth, limiting their contribution to the field. Since the paper is presented as an outcome of a workshop, I suggest reframing and resubmitting it as a perspective article. The focus could center on the key messages from Section 5, which are insightful, while omitting much of the content from Sections 2, 3, and 4, which require significant strengthening.

Detailed Comments

Section 1

The introduction does not align with the structure typically expected of a review article. Instead, it positions the manuscript as a workshop summary. For this reason, it would be more appropriate to present the paper as a perspective rather than a review.

Section 2: Uncertainties in Sea-Level Projections

While this section provides relevant information, the justifications and discussions are superficial. Here are three specific examples:

Lines 136–151: The classification of uncertainties presented here is one approach, but not the only one. A major limitation is the assumption that the boundaries of quantifiable inherent uncertainties can be precisely defined. In practice, these boundaries are often derived from percentiles in probabilistic projections, a convention rooted in established practices (e.g., those of the IPCC), rather than a robust assessment of quantifiable uncertainties. Although the authors' approach is conceptually interesting, its practical application is unclear.

Lines 154–161: If this is meant to be a review, a more thorough engagement with the literature is necessary. For example, the UNEP Emissions Gap Reports could provide insights into the plausibility of different emission pathways.

Lines 190–205: The discussion on interdisciplinary collaboration seems overly broad. If the intent is to emphasize its importance for delivering actionable information, the paragraph should be reframed for clarity, and the section title adjusted to reflect this focus more precisely.

Section 3: Coastal Impacts

This section omits several important topics, such as human interventions, uncertainties in extreme water levels at the coast, and the role of bathymetric changes. Recent relevant studies, such as Toimil et al. (2021, https://doi.org/10.1016/j.earscirev.2020.103110), could be used to strengthen this section. The brief mention of “sediment modeling” (likely referring to sediment transport modeling) overlooks the literature on reduced or appropriate complexity modeling. These studies acknowledge that uncertainties in sediment transport formulas are unlikely to be significantly reduced, but macroscale outcomes can still be modeled. However, these approaches are not yet operational. Overall, this section requires substantial additional work if it is kept in the final version of this manuscript.

Section 4: Uncertainties in Adaptation

This section does not meet the expectations set by its title. Instead of exploring uncertainties in adaptation decisions (e.g., institutional and social dynamics), it focuses primarily on uncertainties in adaptation planning, which are closely tied to sea-level projection uncertainties. The framing of this section should be revisited to clarify its scope and strengthen its discussion.

Section 5: Early Warning Systems

This section effectively highlights the need for early warning systems to support adaptive adaptation planning. While it is not structured as a review, it could serve as the basis for a perspective article.

Conclusion

The conclusions are clear. However, the first recommendation is not well supported by the content in Sections 2, 3, and 4.

---

## [Reviewer Report]

The manuscript describes the structure of a workshop that gathered 22 scientists and policymakers working in sea-level rise and coastal adaptation in the Netherlands, and the outcomes from that workshop. These outcomes are organized along 3 disciplinary areas (sea-level projections, hazards and impacts, and adaptation) and 3 pillars (uncertainties, reduction of uncertainties, and interdisciplinary collaboration). The authors outline areas for improved communication across disciplines and offer guidance, including strengthening/expanding early warning systems for impending dangerous sea-level rise and coastal hazards.

As such, the work is well within the scope of this journal (“addressing major real-world challenges by publishing cross-cutting and interdisciplinary research”) and will be broad interest to the sea-level modeling community through the suggestions for how to generate decision-relevant projections, and to the coastal adaptation and policy-making community through the development of signposts/early warning systems.

The manuscript is well-written and easy to read. I have a mixture of big and small comments and suggestions to improve the clarity of the work, so I gave the recommendation of major revisions. However, given that the work is more of a report/synthesis of a workshop and guidance from the workshop, there are no additional simulations to run or analyses to conduct, so I suspect that my suggestions will not take the authors terribly long to address. I expect that this work will be a valuable contribution to the SLR/coastal adaptation and impacts literature.

Major remarks:

L53, L268, and elsewhere - Reduction of uncertainty is presented as the goal. Is it? Further discussion of possible over- or under-confidence in our sea-level projections, robust decision-making, and regret would be useful to the discussion and framing. This is particularly important given the prominence of Reduction of Uncertainties in the 3 main sections of the manuscript.

L520, and elsewhere - It is claimed a few times that the results are applicable to other coastal areas. This should carry some caveats and limitations, shouldn’t it? Based on geology, geography, geophysical processes driving hazards, local politics and decision-making structures, and so forth. Are some pieces specific to the Netherlands case? At L83, possibly elsewhere, a caveat may be useful that notes that there are probably other processes (“changes in…” of Fig 1) or hazards that must be accounted for in other local contexts.

The manuscript would benefit from some discussion of how feedbacks between the three disciplines here affect uncertainties. As it stands, and perhaps a little bit implied by the “linearity” of figure 1a, it feels as if there is an assumption that this is a model chain, flowing from SLR to hazards/impacts, and then to adaptation. But perhaps a revised fig 1a would show a cycle or more dynamic workflow. For example, in the levee effect, improved adaptation reduces risk/risk perception, and more assets are moved into the floodplain, thereby increasing the needs for adaptation. The SLR discipline can be implicated too in terms of changing uncertainties in SLR projections, which in turn influence hazards, risks, and adaptation response.

Minor remarks:

L42-43 - is early warning systems presented here as an adaptation, or a decision-making structure?

L45 and L63 - “integrated view” - it wasn’t immediately clear that this is to integrate sea-level rise, hazards and impacts, and adaptation, though careful reading by an expert and gleaning from context would lead to this conclusion. The sentence structure in these spots could be revised to clarify this.

The discussion of SLR deep uncertainty is nice and appreciated. Fig 2 is evocative of figure 4 from Bakker et al 2017 (https://www.nature.com/articles/s41598-017-04134-5), but the annotation of the window for early warnings is a useful addition. Given the importance of early warning systems in the take-aways of the manuscript, could a bit more explanation be given about why the arrows shown display the window for early warnings? How does that relate to the uncertainties in the SLR projections, for example, or other uncertainties?

L157, and elsewhere in this paragraph - a definition and brief general description of climate tipping points would be useful to some readers

L160-161 - Is this referring specifically to WAIS, or other tipping points and feedbacks?

L163 - Invoking language regarding structural uncertainty or scenario uncertainty may be useful here. Is this implying that the model structural uncertainty can be reduced? A useful reference comparing structural uncertainty to parametric uncertainty is Yoon et al 2023 (https://www.sciencedirect.com/science/article/pii/S019897152300042X). Relatedly, at L246, it isn’t totally clear what sorts of uncertainties are being referred to by “model uncertainties”.

L216 - is coastal retreat referring to people, ecosystems, or something else? That isn’t totally clear here since the rest of this bullet point refers to elements of the ecosystem.

L240 - this sentence is a bit tough to follow, mostly because it is broken up by a line of references. Not necessarily a problem that must be addressed, just my perception.

L306 - But adaptation scenarios and decision-making is going to carry substantial uncertainties as well. Perhaps for the Netherlands this is codified well in laws or policy procedures/guidance, but that is not something that will generalize globally. Even just the socioeconomic data that underlies cost-benefit calculations is subject to uncertainties, and the projections of these socioeconomic factors.

L334 - would benefit from some examples of the socioeconomic challenges and complexities

L349-350 - similarly, would benefit from some examples of maladaptation and lock-ins

Sec 4.2 - doesn’t feel as strongly connected to uncertainties as the other similar subsections in sec 2 and sec 3. I think this is largely related to my remark about a model chain vs representation of feedbacks between the 3 disciplines. As it stands, the discussion of uncertainties in adaptation feels focused mostly on uncertainty that has propagated from SLR and hazards/impacts.

L371 - this argument is compelling and I like it

L407 - in a couple places, the focus and discussion of early warning systems seems to come out of nowhere. Or rather, if such systems are supposed to be considered a promising adaptation avenue, just one out of many, then it is a bit jarring that we only hear about that one and not any of the other potential adaptation structures that could be considered.

L433 - relatedly, is there any value to co-occurrences of early warnings being triggered? Or this might be another area for further exploration, to see what is the marginal value that information provides.

Figure 3 - the 3 Interdisciplinary Collaboration annotations seem to imply that (say) Sec 2.3 and 3.3 are focused only on the intersection of SLR and Hazards/Impacts, and do not include (say) the intersection of Hazards/Impacts with Adaptation. Is this supposed to be the case?

L457 - “Section 5.1” - that’s the current section - is that right?

L460 - This paragraph makes a great point and call to action for future work.

---

## [Reviewer Report]

General Comments:

This paper presents the results from a workshop held in 2024 in The Netherlands, that was held with 22 scientists and policymakers on the topic of sea-level rise and adaptation. It summarises the results for three research fields (sea-level projections; hazards and impacts; and adaptation) and draws conclusions on research needs and enhanced collaboration. Overall, the paper is interesting and a good addition to the existing literature, as it documents current thinking in The Netherlands from a key group of scientists and also holds useful information for other coastal regions in the world.

However, I found a couple of weaknesses in the scope of the paper, and I also have some detailed comments on the manuscript, that I list below.

My key comment is that the paper and also the set of authors approach the problem of adaptation decision-making from a rather technical-engineering angle. It seems that especially the socio-economic sciences are really cut short here. The disciplinary background of (almost all) of the authors is in natural sciences and engineering. There are now several papers that stress the importance of social science to study aspects of institutional capacities and limits, as important determinants of adaptation actions. In fact several of those studies are mentioned in the paper and appendix (such as Aerts et al. 2024; Barnett et al. 2015), but their importance is really underdeveloped in this paper. This may be due to the participants of the workshop, or because of the lack of science on this in The Netherlands (or maybe both). These dimensions also hold considerable uncertainties for barriers and constraints to adaptation action and require more research efforts and collaboration between the disciplines.

Here, I mean that factors such as justice and equity, as well as social and political limits are essential if we want to understand limits to the rates of SLR that we can adapt to. See for instance the framework proposed by Juhola et al. (2024), as well as the capacities of institutions and actors that need to undertake adaptation actions, and at a more individual level their self-efficacy and outcome efficacy, see Van Valkengoed et al. (2023) that is also cited by the authors.

In sum, I would urge the authors to better integrate the social and institutional dimensions of adaptation, as the current paper provides a rather linear perspective on the topic. At least I got the impression that the authors suggest that if we reduce uncertainties related to SLR projections and hazards and impacts, adaptation action will follow. This is certainly not the case, as rates of change as well as institutional settings are crucial, as underlined by many studies (including the ones quoted above).

Below I provide some further comments and suggestions, that I hope are useful for improving the paper.

Specific comments:

Line 33: Please provide a list of the background and disciplines (not necessarily names) of the workshop participants. There are 18 authors, so 4 people are missing. Also, who were the policymakers? Also, a comment on the balance between science and policy participation could be made…

Line 41: The topics described in this paper (SLR, hazards and impacts, and adaptation) are not disciplines. They are themes, that can be studied by different research disciplines. Please adjust the language here and in other locations. Also, what disciplines are implied in this paper? These never appear anywhere. It strikes me that social science disciplines such as psychology and behavioural science, economics, anthropology, political science are not mentioned. Only the aggregate term “socio-economic” is used in several places, but this is unspecific and does not do justice to the complexity and uncertainties (and knowledge) that exist in each of these fields. See also my general comment, above.

Lines 95-96: This is an important observation. Maybe it could be repeated in the abstract, and please also as mention options to improve this interaction in Section 5.

Lines 97-102: These are (largely) natural science conferences as far as I can tell. Can you please also mention social science/economic science conferences where these topics would/could be discussed? Or are these disciplines included in these two conferences?

Lines 133-134: But what seems to be implied is that the mean SLR also results in higher extreme sea-levels (as also IPCC concludes). This is a bit misleading, as if mean sea-level is the real threat for flooding, maybe some other wording can be found? Mean sea-level translates into both changes for salt-intrusion and morphologic changes, as well as extreme sea-level events during storm surges.

Line 140: Probably, you mean plural (models) here. As this is true for the GCMs but also for the ice sheet and sea-level models that depend on projections from those.

Lines 146-147: Here, it seems as if AMOC and collapse of the WAIS work on the same timescales. But the meters from WAIS are probably on a longer timescale. Please clarify and make this text consistent.

Lines 148-151: I would suggest that this text precedes the text starting on Line 144. This makes the flow more logic for the reader.

Line 152, Figure 2: Here I have two comments: 1) What does the “Window for early warnings” indicate precisely? Why has it this length? Would the window not end when the timing of the star is reached? This is unclear and not described in the text. 2) Is the trajectory where the sea-level returns to the “normal pathway” at number 3 realistic? What mechanism would cause this? Please explain.

Line 168: Please replace the word “new” with “previously”.

Lines 178-180: This paper was written in 2016 and published in 2017, now almost 9 years ago. Is this estimate of “decades” still realistic? Given also that machine learning is rapidly improving model parametrisations and also process modelling, I wonder if we are really still decades away from having higher resolution modelling. Finally, I also wonder how model resolution compares to fundamental process understanding (not explicitly mentioned here) and lack of modern-day observations of ice sheet instabilities for instance, when it comes to limits in our modelling capabilities.

Line 188: Probably you mean the “current” and not “upcoming” CMIP7.

Lines 201-203: Please include here also the paper by Katsman et al. (2011) that documents the work done for the Delta Committee in 2008, which was essential in pushing high-end scenarios for policymaking in The Netherlands.

Line 238: Please remove the word “data”. This is really much more than data issues, as also fundamental understanding of vulnerability (including in models, and scoail vulnerability) is of importance here.

Line 252: Please add here the word “river” to discharge regimes.

Lines 276-277: It seems to me that elevation data is no issue in Netherlands. Either cut, or make a statement on other world regions.

Line 378: Please add costs.

Line 381: At this point, a discussion of limits would be needed, related to social and institutional capacities and barriers. A key question in this regard, with all the research carried out in The Netherlands in the last few years; have these social, economic and institutional aspects been (sufficiently) addressed?

Line 390: If these are projections, the word “will” does not make sense. Then please replace with the word “could”.

Lines 428-429: Again, here there seems to be some linear thinking. Early warning systems are not effective only when uncertainties about future changes are reduced. They can only be effective when institutions can act effectively upon receiving these signals … please see my first point about limits (and constraints/barriers). Please reword this sentence accordingly, or add a perspective about the possible use of this information and capacities of institutions to adapt.

References:

Aerts, J.C.J.H. et al. 2024. Nature Water 2, 719-728. https://doi.org/10.1038/s44221-024-00274-x

Barnett, J. et al. 2015. Ecology and Society 20(3), 5. http://dx.doi.org/10.5751/ES-07698-200305

Juhola, S. et al. 2024. Global Environmental Change, 87, 102884. https://doi.org/10.1016/j.gloenvcha.2024.102884

Katsman, C.A. et al. 2011. Climatic Change 109, 617-645. https://doi.org/10.1007/s10584-011-0037-5

Van Valkengoed, A.M. et al. 2023. Risk Analysis, 44, 553-565. https://doi.org/10.1111/risa.14193

---

## [Editor Report]

Dear Authors, your paper has been comprehensively reviewed by three excellent reviewers. Three reviewers recognised the value of the paper, with two recommending publication after major revision. The recommendation of the third reviewer is to consider the resubmission of the work as a perspective paper. This should be considered but I am not suggesting this as a requirement.

Please take your time and respond to all reviewer questions and suggestions.

---

## [Reviewer Report]

I read the response of the authors and the new manuscript. The manuscript has been imporoved significantly,; both in terms of content and scoping. Given that the authors now clearly say that this is the outcome of a workshop and based on a Dutch perspective, I have no objections to the paper being published.

I have only one minor additional comment for consideration by the authors:

- lines 191 - 197: the concept of tipping point is used here, but crossing or having crossed a tipping point does not mean that abrupt changes will occure. I suggest clarifying.

---

## [Reviewer Report]

Dear Authors, I appreciate the attention paid to my comments; all comments have been carefully addressed and I think the current version of the paper can be accepted for publications.

---

## [Editor Report]

The authors are thanked for the effort to produce an amended version of the manuscript. There is one minor comment from one of the reviewers that can be considered. I would not be in favour of returning the manuscript for revision for this comment only. Other than that, the manuscript can be accepted for publication.